# Growth and Characterization of Carbon Nanofibers Grown on Vertically Aligned InAs Nanowires via Chemical Vapour Deposition

**DOI:** 10.3390/nano13243083

**Published:** 2023-12-05

**Authors:** Muhammad Arshad, Lucia Sorba, Petra Rudolf, Cinzia Cepek

**Affiliations:** 1Istituto Officina dei Materiali—CNR, Laboratorio TASC Area Science Park—Basovizza, Edificio MM, Strada Statale 14, Km.163.5, I-34149 Trieste, Italy; muhammad.arshad@ncp.edu.pk; 2Zernike Institute for Advanced Materials, University of Groningen, Nijenborgh 4, NL-9747AG Groningen, The Netherlands; p.rudolf@rug.nl; 3Nanosciences and Technology Department, National Center for Physics, Quaid-i-Azam University Campus, Islamabad 2141, Pakistan; 4NEST, Istituto Nanoscienze-CNR and Scuola Normale Superiore, Piazza S. Silvestro 12, I-56127 Pisa, Italy; lucia.sorba@nano.cnr.it

**Keywords:** carbon nanofibers, indium arsenide nanowires, hybrid nanostructures, chemical vapor deposition, interconnects

## Abstract

The integration of carbon nanostructures with semiconductor nanowires holds significant potential for energy-efficient integrated circuits. However, achieving precise control over the positioning and stability of these interconnections poses a major challenge. This study presents a method for the controlled growth of carbon nanofibers (CNFs) on vertically aligned indium arsenide (InAs) nanowires. The CNF/InAs hybrid structures, synthesized using chemical vapor deposition (CVD), were successfully produced without compromising the morphology of the pristine nanowires. Under optimized conditions, preferential growth of the carbon nanofibers in the direction perpendicular to the InAs nanowires was observed. Moreover, when the CVD process employed iron as a catalyst, an increased growth rate was achieved. With and without the presence of iron, carbon nanofibers nucleate preferentially on the top of the InAs nanowires, indicating a tip growth mechanism presumably catalysed by a gold-indium alloy that selectively forms in that region. These results represent a compelling example of controlled interconnections between adjacent InAs nanowires formed by carbon fibers.

## 1. Introduction

Carbon nanotubes (CNTs) and carbon nanofibers (CNFs) possess exceptional mechanical and electrical properties, making them promising materials for interconnecting wires in future very large-scale integration technology [1,2]. Simulations have indicated that the use of metallic CNT interconnects could lead to more energy-efficient and faster integrated circuits [3]. However, achieving precise control over the interconnection of individual nanowires (NWs) with CNFs/CNTs remains a significant challenge. One promising approach for integrating CNTs into real devices is the synthesis of controlled CNT/NW hybrid nanostructures. The possibility of obtaining well-defined CNT/metal or semiconducting NW hybrid structures has been widely explored in the literature [4] but controlled growth for practical electronic devices has not yet been realized. Semiconducting nanowires (NWs) can be grown in ordered and oriented arrays on semiconducting substrates using chemical beam epitaxy (CBE), and these precisely ordered one-dimensional nanostructures can serve as templates for synthesizing CNTs/CNFs and obtaining the desired hybrid structures [5]. Among the various techniques for CNT/CNF growth, catalytic chemical vapor deposition (CVD) holds the most promise due to its ability to control the location and diameter of the tubular structures, which is determined by the position and size of the catalyst nanoparticles. Moreover, the relatively low growth temperatures (400–900 °C) involved in CVD enable direct deposition onto electronic devices [6,7]. CVD, therefore, provides a cost-effective and straightforward means of synthesizing patterned building blocks for large-scale hybrid nanostructure fabrication. 

In this study, we conducted CVD growth using C_2_H_2_ as the precursor gas on a network of vertically aligned InAs NWs, demonstrating the synthesis of CNFs without compromising the integrity of the pristine NW ensemble. Generally, the CVD growth of CNTs/CNFs requires transition metal (Ni, Fe, Co) catalysts to decompose the gaseous carbonaceous precursor [6]. Hence, we also investigated growth in the presence of Fe as a catalyst. The combination of an in situ X-ray photoelectron spectroscopy (XPS) study, ex situ characterization using scanning electron microscopy (SEM), and Raman spectroscopy allowed for a comprehensive analysis of the samples. These findings contribute to our understanding of how to realize controlled interconnections between adjacent nanowires with carbon fibers and hold great potential for the application of such hybrid nanostructures in energy-efficient integrated circuits and nanoscale devices.

## 2. Materials and Methods

### 2.1. Growth of the Indium Arsenide Nanowire Arrays

The vertical InAs NWs used in these experiments were fabricated at the Istituto Nanoscienze-CNR (Pisa, Italy). The InAs NWs have grown on an InAs(111)B substrate using chemical beam epitaxy (CBE) as detailed in ref. [8]. InAs substrates coated with a 0.1 nm thick Au film were introduced into the CBE chamber; pre-growth thermal treatments were performed in situ in the CBE chamber. The growth of all NWs followed the same protocol: (i) a temperature ramp was applied, starting from the standby temperature (~250 °C) and gradually increasing to the annealing temperature (520 ± 10 °C), while maintaining a tertiarybutylarsine (TBAs, Dockweiler Chemicals GmbH, Marburg, Germany, epigrade) line pressure of 5.3 mbar. The annealing step ensured the formation of Au nanoparticles through the thermal dewetting of the film and facilitated the desorption of surface oxide from the InAs substrate. (ii) Following the annealing step, the temperature was gradually decreased from the annealing temperature to a growth temperature of 430 ± 10 °C. (iii) The growth duration was 45 min, maintaining a line pressure of 0.4 mbar and 2.7 mbar for trimethylindium (TMIn, Dockweiler Chemicals GmbH, Marburg, Germany, optoelectronic grade) and TBAs, respectively. (iv) Subsequently, the temperature was lowered to ~250 °C without the presence of TMIn, while maintaining a linearly decreasing TBAs line pressure [8]. Transport to Trieste implied exposure to air.

### 2.2. Chemical Vapor Deposition of the Carbon Nanofibers

CVD growth and XPS analysis were performed in the INSPECT laboratory of the TASC-IOM-CNR institute, where all steps of the CVD process, including catalyst deposition, can be carried out in an ultra-high vacuum experimental setup (base pressure < 1 × 10^−10^ mbar). The typical CVD growth protocol comprised preliminary out-gassing of the InAs substrate at 300–400 °C using a home-made silicon heater, followed by H_2_ (SIAD, Bergamo, Italy, grade 5) exposure to pre-treat and clean the surface (*vide infra*); the pressure in the growth system during this treatment was ~4 × 10^−4^ mbar. The temperature was measured using an infrared pyrometer. The typical duration of these pre-treatments was 15 min. Subsequently, the sample was annealed to the growth temperature (chosen in the range 490–550 °C) and CVD was performed with and without a catalyst (iron), typically for 25–40 min with C_2_H_2_ (SIAD, Bergamo, Italy, grade 2.5) and H_2_ (SIAD, Bergamo, Italy, grade 5) as precursor gasses in the flux range 0.5–7.0 sccm; the pressure in the growth chamber during the process was in the range of 10^−4^ mbar. When a catalyst was used, Fe films were deposited using electron bombardment (Fe target from Sigma-Aldrich, St. Louis, MI, USA, 99.9% purity) onto the InAs substrate kept at room temperature at a growth rate of ~0.35 Å/min before starting CVD. The Fe deposition rate was obtained from the attenuation of the photoemission peaks of the In3*d* core level and confirmed using a home-made quartz microbalance. Catalyst deposition was followed by annealing to the chosen growth temperature. 

### 2.3. Surface Characterization

X-ray photoelectron spectroscopy: The growth chamber is directly connected with the XPS analysis chamber, equipped with a conventional non-monochromatized MgKα X-ray source (hν = 1253.6 eV) and a 120° hemispherical electron energy analyser (RESOLVE 120, PSP Vacuum Technology, Macclesfield, UK). Thus, it was possible to control the chemical state of the substrate before and after the growth via XPS and to precisely monitor the influence of all CVD growth parameters. The XPS spectra were acquired before and after all CVD steps in normal emission geometry, with an energy resolution of 0.8 eV. Analysis of the data was conducted by performing a non-linear mean square fit, reproducing the photoemission intensity using Doniach-Sunjic line shapes superimposed onto a Shirley background. Binding energies were calibrated by fixing the C1*s* binding energy of adventitious carbon to 284.6 eV [9]. Binding energy (BE) positions are given with a precision of ±0.1 eV. 

Scanning Electron Microscopy: SEM observations were performed using a Field Emission In-lens SEM (JSM 890, JEOL, Basiglio, MI, Italy) at 10 kV accelerating voltage and a 1 × 10^−12^ A probe current, delivering a spatial resolution of 1 nm. Images were obtained with both secondary electron and back-scattered signals. The instrument is equipped with an X-Act detector (Oxford Instruments) for energy dispersive X-ray spectroscopy (EDS).

Raman spectroscopy: Raman scattering measurements were performed with a μ-Raman spectrometer (Renishaw 1000, Wotton-under-Edge, UK) using the 514.5 nm line of an Ar^+^ laser as an excitation source with a spot diameter of about 1 μm. All measurements were performed at room temperature in backscattering geometry, with the wires in the plane of incidence of the incoming light.

## 3. Results and Discussion

### 3.1. Substrate Pre-Treatment

Scanning electron micrographs of Au-catalysed InAs NWs grown using chemical beam epitaxy on an InAs(111)B substrate are presented in Figure 1a,c and show vertically aligned nanowires of different diameters, which randomly cover the substrate surface. The length of the InAs nanowires seems different after H_2_ pre-treatment because the SEM images are collected at different tilt angles. Energy-dispersive X-ray spectroscopy (not shown) performed together with scanning electron microscopy revealed that the Au nanoparticles, used as catalysts, reside at the tips of the InAs NWs. Note that the density of InAs NWs varies from sample to sample; when discussing the CNF growth, we shall, therefore, refer to the density of CNFs/InAs nanowire, expressed as a percentage. We tested the thermal stability of the InAs NWs to annealing in UHV at increasing temperatures using the same annealing time as typically used in the CVD process (≈25 min). The SEM images in Figure 1 demonstrate that no melting or significant morphological changes occur up to ~520 °C when annealing without (Figure 1b) or with H_2_ (Figure 1d), annealing at a temperature higher than 530 °C partially destroys the NWs (note that the InAs melting temperature in bulk form is 942 °C [10] and the InAs NW growth temperature is 420–440 °C). The H_2_ pre-treatment is essential because when CVD growth (*vide infra*) was performed without this pre-treatment, we never observed the growth of tubular carbon nanostructures, and the SEM images were similar to those of the as-received samples (not shown). This is most probably due to the oxide layer formed on the whole surface of the substrate and of the NWs after air exposure, which does not support any CNF nucleation within the CVD parameter window we used. The optimized procedure before CVD was, therefore, the following: the InAs NWs were first annealed at ~430 °C for 10 min, then underwent the hydrogen pre-treatment at ~520 °C using a flux of 3 sccm for 15 min (pressure during the treatment: ∼4 × 10^−4^ mbar). 

To evaluate whether the cleaning procedure was efficient in removing the oxide layer formed by exposure to air, the chemical effects of these treatments were studied using X-ray photoelectron spectroscopy. Figure 2 shows the As3*d* and In3*d* XPS spectra of the as-received sample, after annealing at 430 °C, 500 °C and after H_2_ pre-treatment at 525 °C. Both the In3*d* and As3*d* core level spectra of the as-received sample (Figure 2a,b bottom) are fitted with two components; the more intense one corresponds to the In-As bond (In3*d*_5/2_ peaked at a BE of ≈444.6 eV, As3*d* at ≈41.0 eV, marked in blue). The other component in each spectrum is associated with the native surface oxide (marked in green), composed of In_2_O_3_ (In3*d*_3/2_ component peaked at a BE of ≈445.5 eV) and of a mixture of As_2_O_3_ and As_2_O_5_, giving rise to the As3*d* component peaked at a BE of ≈45.6 eV [11,12]. The weak peak at about 443 eV in the In3*d* spectrum (marked in orange) is due to the MgKα satellites. 

Annealing at 430 °C leads to the partial desorption of In oxide and the complete desorption of the As oxide (Figure 2a,b second spectra from bottom), as well as causing partial desorption of adventitious carbon (data not shown). Annealing at 500 °C and H_2_ pre-treatment at 525 °C result in a slight further desorption of carbon and also of oxygen. The desorption of the oxides (Figure 2a,b top spectra) goes hand in hand with a change in the ratio between the As3*d* and In3*d* intensities. While the as-received InAs NWs are stoichiometric within the experimental error, annealing at high temperatures causes the partial desorption of As, with the extent increasing as the temperature rises (Figure 2a, inset). It is worth noting that the photoelectron escape depth in our experimental conditions is approximately 25 Å for As and 18 Å for In [13]. Therefore, As desorption occurs at least within the topmost 2–3 nm but does not affect the core of the NWs, as evidenced by the Raman spectra discussed below. Arsenic desorption after annealing in UHV is a well-known phenomenon resulting from the partial decomposition of InAs. The temperature and rate of arsenic desorption depend on the structure of the InAs NWs [14].

### 3.2. CVD Synthesis

#### 3.2.1. Scanning Electron Microscopy Results

Synthesis of tubular carbon was successful only on substrates that underwent H_2_ pre-treatment. CVD growth was carried out with and without the use of iron as a catalyst, employing various H_2_ and C_2_H_2_ flow rates ranging from 0.5 sccm to 7.0 sccm. The growth temperatures were within the range of 470–550 °C, i.e., in a temperature window where the density, orientation, and length of the InAs NWs remained nearly unchanged after annealing in ultra-high vacuum (Figure 1) and beyond. When the growth was attempted at temperatures below approximately 490 °C, no significant changes compared to the as-grown InAs substrate were observed. This suggests that these temperatures are insufficient to decompose C_2_H_2_ on InAs NWs in our experimental conditions. The formation of new structures at the tips of the nanowires was observed within the temperature window of 500–530 °C. Figure 3a shows the SEM image acquired after CVD growth performed without catalyst at ~525 °C using a flux of ~7 sccm of C_2_H_2_ for 20 min (pressure in the preparation chamber during growth: ~8 × 10^−4^ mbar). In this case, a deposit of almost spherical nanoparticles is visible on both the InAs NWs and the InAs substrate, together with short tubular structures mainly located at the NW tips. As seen in the SEM micrograph shown in Figure 3b, the addition of H_2_ to C_2_H_2_ during CVD helped to nearly eliminate the presence of the spherical carbon nanoparticles and to enhance the synthesis of CNFs. In this case, we used an acetylene flux of 3.6 sccm together with a hydrogen flux of 2 sccm at 525 °C (pressure during growth ~4.5 × 10^−4^ mbar). We observed that in the appropriate growth conditions, these tubular structures have diameters of a few nanometers, nucleate preferentially at the InAs NW tips, grow along the direction perpendicular to the NW axis, and sometimes connect two NWs. In detailed SEM micrographs of the sample of Figure 3b, which are shown in Figure 4a,b, one clearly distinguishes the nucleation of CNFs at the NW tips (a) and the establishment of a CNF bridge (b) between two InAs NWs. That these tubular structures are indeed carbon nanofibers is demonstrated by the Raman spectra discussed below. 

In our attempt to optimize the growth parameters, we explored whether increasing the C_2_H_2_ flux, the H_2_ flux, the growth time, and temperature could increase the density of CNFs produced. The results are shown in Figure 5. We found that the number of CNFs per InAs NW depends strongly on the growth conditions; in particular, first, we established, as illustrated in Figure 5a for a H_2_ flux of 2 sccm, that a C_2_H_2_ flux of 3.6 sccm was optimal; we established that the maximum percentage of CNFs per InAs NW without Fe catalyst obtainable in our parameter window was around 15% (Figure 5a blue dots), while it reached 50% with the iron catalyst (Figure 5a red dots). The impact of growth temperature on the number of CNFs per InAs NW with and without the iron catalyst was explored as well and the results are summarized in Figure 5b. Notably, at temperatures below the melting point of the InAs NWs, the density of CNFs increases, reaching 50% when the catalyst is present (Figure 5b red dots), which is three times higher than that without the use of an iron catalyst (Figure 5b blue dots), while the tube diameter decreases. When attempting CVD growth at temperatures of 530 °C or higher, the SEM images (see for ex. Figure 3d) revealed partial melting of InAs nanowires, consistent with previous literature findings [15]. At temperatures exceeding 540 °C, no tubular structures were observed under any tested conditions, and the SEM images showed partially or completely melted InAs NWs (not shown). 

Figure 6 further illustrates CVD growth at the optimal conditions 525 °C, C_2_H_2_ flux 3.6 sccm and H_2_ flux 2.0 sccm) with and without the iron catalyst. It is evident that the InAs NWs undergo shape changes, becoming approximately 15% to 20% shorter in length compared to before the growth, indicating partial melting. As seen in the SEM micrograph in Figure 6a, which refers to the sample shown in Figure 3c but on a larger scale, depositing a thin film of iron (≈0.6 nm) before CVD growth causes a strong increase in the density of the tubular structures, which are also significantly longer and thinner than those obtained when growing without Fe (Figure 6b).

#### 3.2.2. Spectroscopic Characterization

To gain a better understanding of the structures grown by CVD and the chemical changes induced by the growth process, Raman and XPS spectra were collected. While we have referred to the resulting structures as carbon nanofibers based on the CVD growth, the confirmation comes solely from the Raman data. 

Figure 7 displays the Raman spectra acquired on the InAs substrate covered with the as-received InAs NWs, after annealing at 430 °C, and after CVD growth as described in Section 2.2 above, namely approximately 4.5 × 10^−4^ mbar of C_2_H_2_ (3.6 sccm) and H_2_ (2.0 sccm) for 40 min at 525 °C, without Fe catalyst. This sample is the one also shown in Figure 3b. The peaks at 212 cm^−1^ and 237 cm^−1^ (Figure 7, left panel) correspond to the typical tangential optical (TO) and longitudinal optical (LO) phonon modes of the InAs NWs [16]. These spectra show no significant differences before and after growth, indicating that the NWs remain largely unaffected by the various processes. No additional peaks appeared in this spectral region after CVD growth, ruling out the production of single-walled carbon nano- tubes (CNTs) under our growth conditions. In fact, if single-walled CNTs were formed, one would expect to observe the radial breathing mode (RBM) signature in this spectral region [17,18].

At higher wavenumbers, the spectrum acquired after CVD growth (Figure 7, right panel) exhibits a G peak at 1592 cm^−1^, which is characteristic of sp^2^ carbon, and a strong D peak at 1368 cm^−1^, indicative of defects and disordered graphitic material [19,20,21,22,23], Both the D and G peaks are broad, suggesting the presence of disordered carbon and non-crystalline structures, such as carbon nanofibers and amorphous nanoparticles [24,25,26]. The intensity ratio I_D_/I_G_ informs qualitatively on how ordered a carbon material is [27] and the high I_D_/I_G_ ratio of 0.75, deduced from the spectrum in Figure 7, corresponds to a low degree of order typical for CNFs [28]. No significant changes in the Raman spectra were observed when the CVD growth was performed in the presence of the Fe catalyst (data not shown), indicating that in all cases, the carbon structures are similar and highly disordered.

In all instances (with and without H_2_ during CVD growth, and with and without deposition of Fe), CNFs preferentially nucleate at the tips of the InAs NWs, where the Au nanoparticles used as catalysts for NW growth are located [29]. This suggests that the tips play a crucial role in CNF synthesis. It should be noted that if CNF growth were catalysed by In, As, and/or Fe (when used), one would expect to observe CNF nucleation on the substrate surface as well, along with potential chemical interactions between In or As and carbon, which could be detected using XPS. Additionally, the formation of iron compounds with indium and/or arsenide is unlikely at our growth temperatures [30,31].

In order to attempt to gain insights into the nucleation and growth process via the bonds formed, we acquired XPS spectra of the 3*d* core levels of In and As, as well as of the C1*s* core level region before and after CVD growth; the results are presented in Figure 8. The C1*s* spectrum (Figure 8a) obtained after CVD growth reveals a sharp intensity decrease in the component at approximately 287.2 eV in binding energy (green component), primarily attributed to (C-O_x_) contaminants that were apparently largely etched away during the CVD process. Furthermore, there is a decrease in the component at a BE of approximately 285.2 eV (white component), attributed to disordered and/or sp^3^ carbon, and an increase in the component related to sp^2^ bonds at approximately 284.4 eV BE (blue component) due to fiber nucleation. Meanwhile, the As3*d* spectrum after CVD growth (Figure 8b, top) shows the appearance of a component at a BE of 43.4 eV (green component), corresponding to As^+1^ [32]. This suggests the formation of As-C bonds [33,34]. It is already known that Au nanoparticles can catalyse the decomposition of InAs during high annealing in UHV [14,35]. Arsenic exhibits a low solubility in Au and at high temperatures, it sublimates in the form of As_x_. Therefore, during the CVD process, it can react with C_2_H_2_ to form As carbide, as revealed by our XPS data. On the other hand, Indium has a high solubility in Au and readily forms Au-In alloys [36]. Unfortunately, due to the low Au concentration in our sample, we were not able to distinguish the formation of an In-Au alloy, and the In3*d* spectra acquired before and after the CVD process (Figure 8c) did not show any significant differences. Nevertheless, it is worth noting that the AuIn_2_ alloy in the form of nanoparticles has previously been found to catalyze the growth of InAs nanotrees even at very low temperatures [29] and it may be responsible for the CNF synthesis in our case. The nanometric size of the Au-In wires in the samples studied here can further enhance their reactivity, as already observed in several nanostructured materials [37,38,39,40].

## 4. Conclusions

As summarized in the scheme in Figure 9, carbon nanofiber-InAs hybrid nanostructures were successfully synthesized by CVD at optimized growth conditions. SEM micrographs confirmed that the original ensemble of InAs nanowires is thermally stable (preserved) during annealing, H_2_ pre-treatment, and CVD growth of carbon nanofibers.

When the growth was performed only with acetylene, carbon nanoparticles (C NPs) were found on the surface and on the InAs nanowires; C NPs were absent when hydrogen was added. In the latter conditions, Raman spectroscopy revealed the presence of graphitic-like carbon structures with a high number of defects, which points to carbon nanofibers. The carbon nanofibers preferentially nucleate at the tip of InAs nanowires, probably catalyzed by the gold-indium alloy, which can form only there. The number density of CNFs increased from 15% to 50% of decorated InAs nanowires when Fe was added as a catalyst in the CVD process. Our results demonstrate that interconnections between adjacent InAs NWs with carbon fibers can be obtained via CVD; however, to which of the neighboring NWs a connection is made, is random.

## Figures and Tables

**Figure 1 nanomaterials-13-03083-f001:**
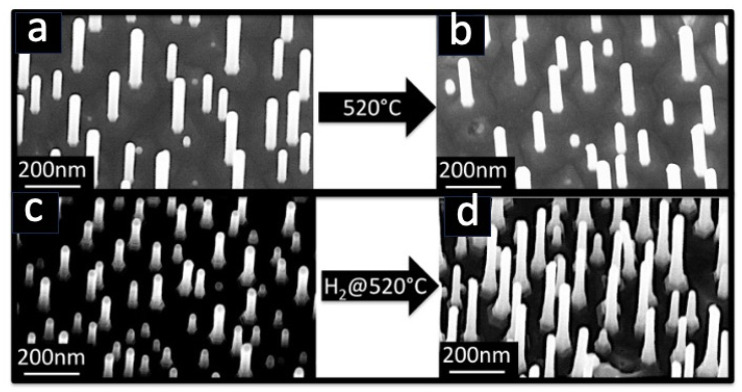
SEM images of as-grown InAs NWs (**a**,**c**) and after pre-treatment of the same sample at 520 °C without (**b**) or with (**c**) 3 sccm of H_2_ (see text for more details). The images (**a**,**b**) were obtained at a tilt angle of 10.0° and the images (**c**,**d**) were acquired at tilt angles of 10.0° and 20.0°, respectively, which explains why the length of the InAs NWs seems different.

**Figure 2 nanomaterials-13-03083-f002:**
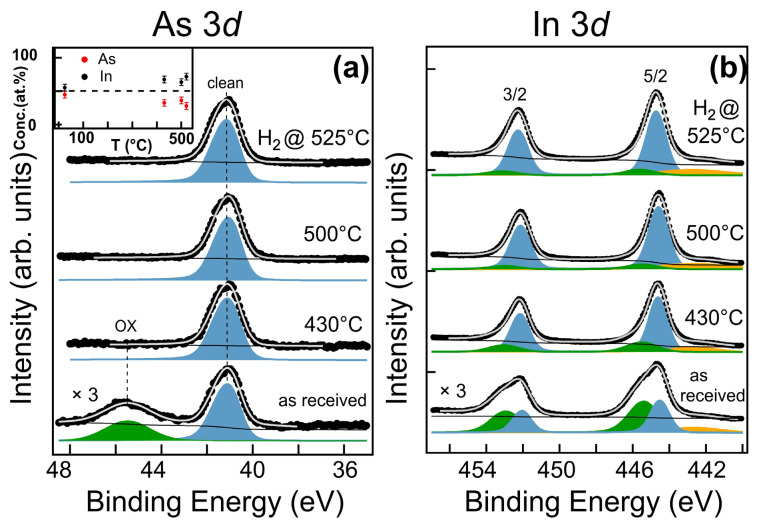
XPS spectra of the As3*d* (**a**) and In3*d* (**b**) core level regions of as-received InAs nanowires (bottom), after annealing at 430 °C, 500 °C and after H_2_ pre-treatment at 525 °C (top) and corresponding fits. The inset shows the As (red) and In (black) concentrations after the same three pre-treatments. The dotted line corresponds to stoichiometric InAs (50%). The green and blue components refer to, respectively, oxidized and non-oxidized InAs-related components, while the yellow component is due to the MgKα satellites (see text for more details).

**Figure 3 nanomaterials-13-03083-f003:**
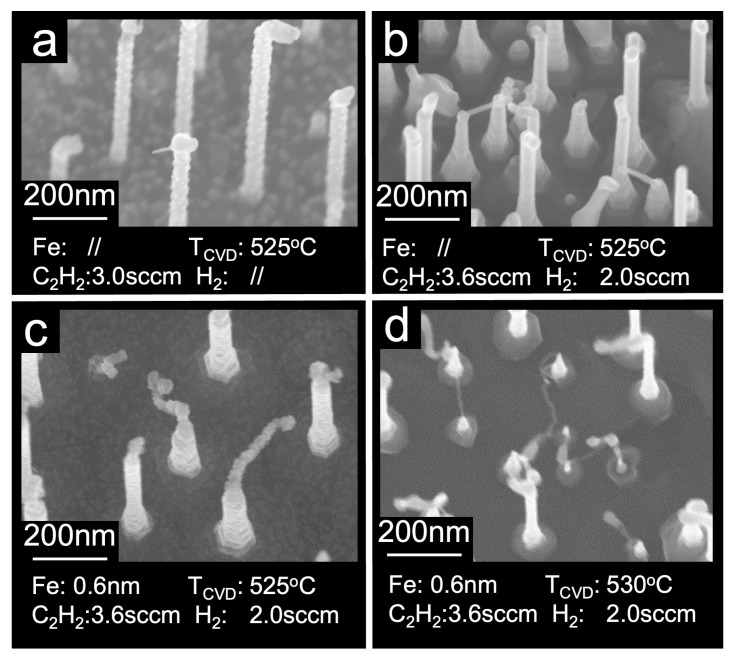
SEM images collected after different CVD processes: (**a**) without catalyst, ~8 × 10^−4^ mbar of C_2_H_2_ (3.0 sccm) for 20 min at ~525 °C; (**b**) without catalyst, ~4.5 × 10^−4^ mbar of C_2_H_2_ (3.6 sccm) and H_2_ (2.0 sccm) for 40 min at 525 °C; (**c**) with catalyst (0.6 nm Fe), same CVD condition as (**b**); (**d**) same conditions as (**c**) but at higher temperature (530 °C), where it is clear that part of NWs are strongly distorted and/or etched (see text for more details). All the images were collected at a tilt angle of 20.0° (**a**,**b**,**d**) and 10.0° (**c**).

**Figure 4 nanomaterials-13-03083-f004:**
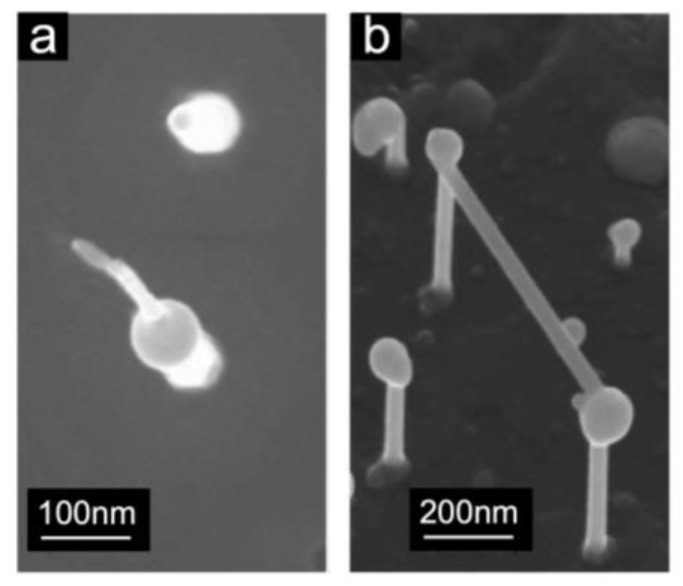
SEM images collected after different CVD processes: (**a**,**b**) are detailed views of the sample shown in Figure 3b grown without catalyst, with ~4.5 × 10^−4^ mbar of C_2_H_2_ (3.6 sccm) and H_2_ (2.0 sccm) for 40 min at 525 °C. The images were collected at a tilt angle of 30.0° (**a**) and 0.0° (**b**).

**Figure 5 nanomaterials-13-03083-f005:**
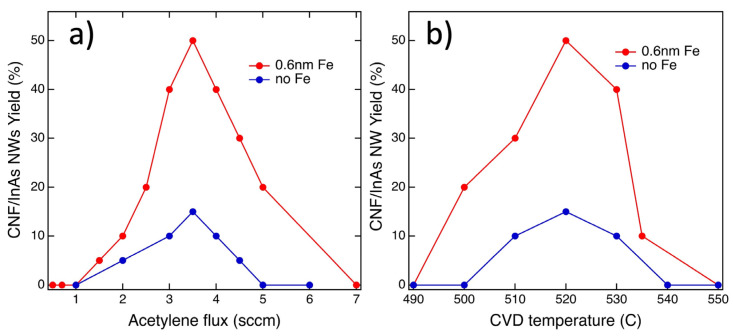
Yield of CNF growth: percentage of CNFs connecting InAs NWs when (**a**) the H_2_ (2.0 sccm) flux is kept constant but the C_2_H_2_ flux is varied during CVD growth at 525 °C with (red dots) and without the iron catalyst (blue dots); (**b**) both the C_2_H_2_ (3.6 sccm) flux and the H_2_ (2.0 sccm) flux are kept constant but the temperature is varied during CVD growth with (red dots) and without the iron catalyst (blue dots).

**Figure 6 nanomaterials-13-03083-f006:**
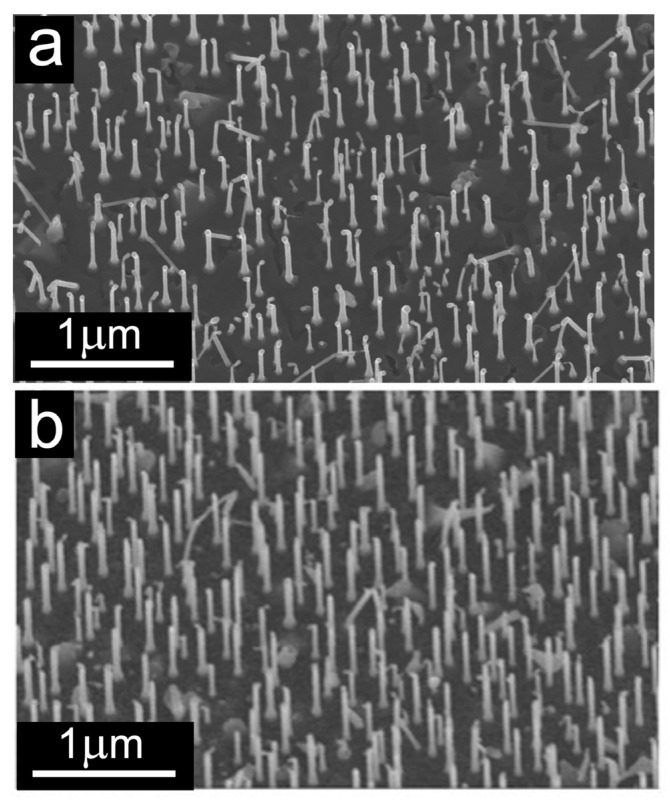
SEM image of (**a**) the sample shown in Figure 3c referring to CVD growth (~4.5 × 10^−4^ mbar of C_2_H_2_ (3.6 sccm) and H_2_ (2.0 sccm) for 40 min) at 525 °C after deposition of 0.6 nm Fe on a larger scale; (**b**) the sample shown in Figure 3b referring to CVD growth under the same conditions but without the catalyst, also on a larger scale. These images were collected with a sample tilt of 20.0°.

**Figure 7 nanomaterials-13-03083-f007:**
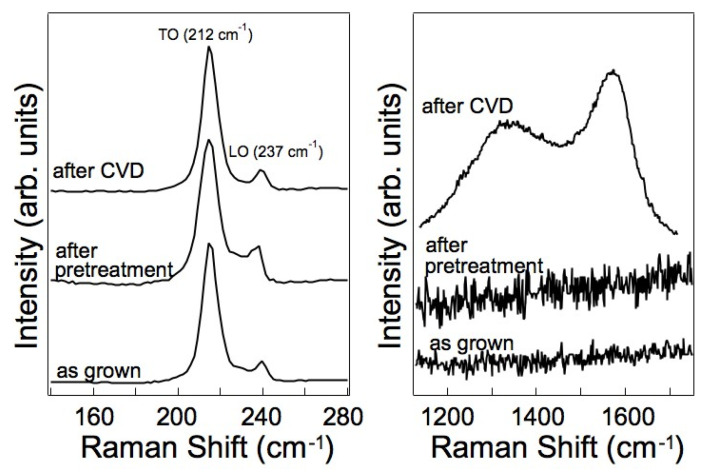
Raman spectra of as-grown InAs NWs, after annealing at 430 °C and after CVD growth resulting in the sample shown in Figure 3b. The left panel shows the low wavenumber region, while the right panel displays the high wavenumber region.

**Figure 8 nanomaterials-13-03083-f008:**
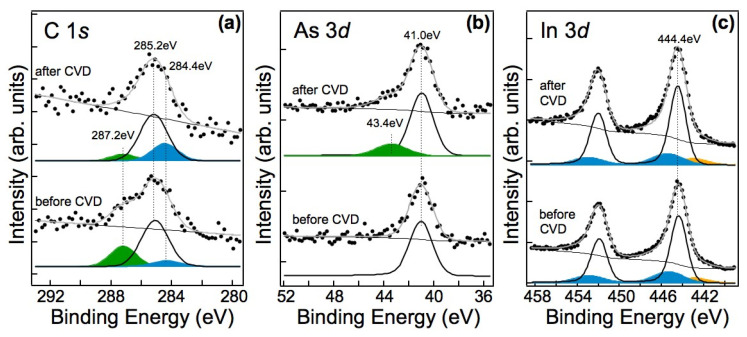
XPS spectra of the (**a**) C1*s*, (**b**) As3*d*, and (**c**) In3*d*, core level regions of recorded in situ on the InAs nanowire array before and after CVD and corresponding fits. The weak peak at about 443 eV in the In3*d* spectra (filled in yellow) is due to the Mg Kα satellites. The attribution of the various components in each spectrum is discussed in the text.

**Figure 9 nanomaterials-13-03083-f009:**
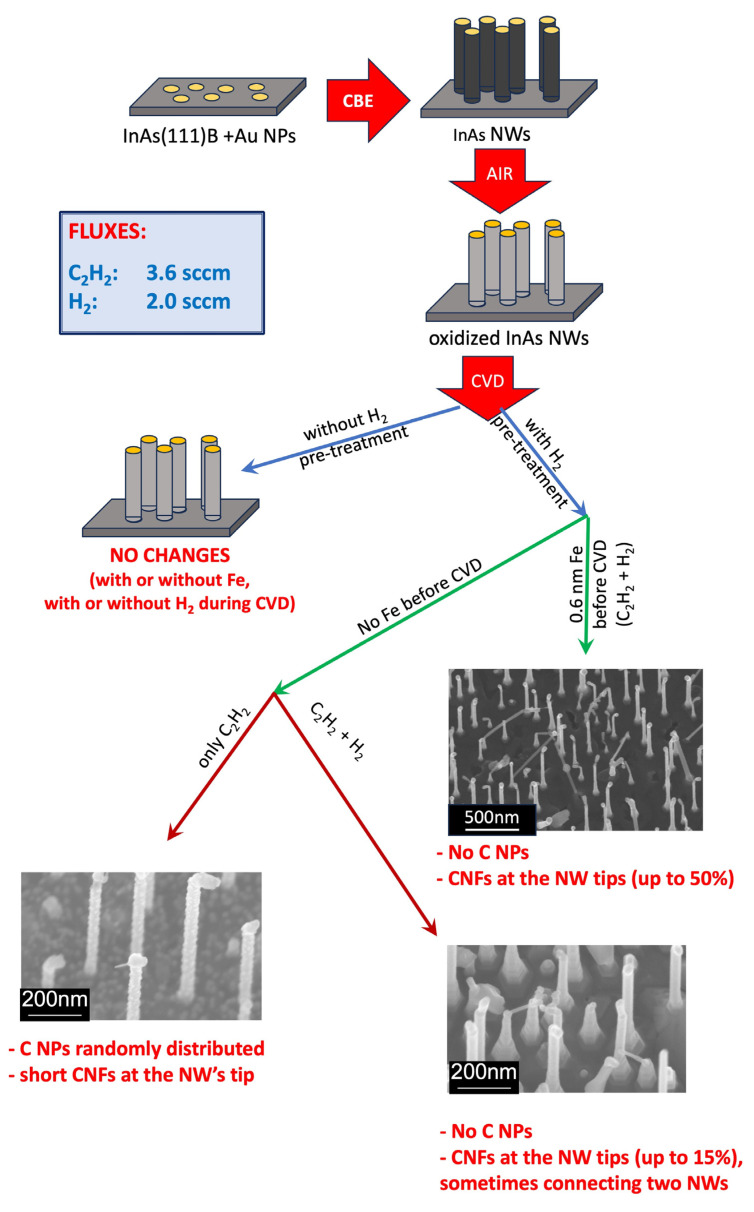
Scheme summarizing the procedures and findings.

## Data Availability

Data are contained within the article.

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
