# Peer review of "Growth and Characterization of Carbon Nanofibers Grown on Vertically Aligned InAs Nanowires via Chemical Vapour Deposition"

_nanomaterials, 2023, doi:10.3390/nano13243083_

Round 1

Reviewer 1 Report

Comments and Suggestions for Authors

Review on “Growth and characterization of carbon nanofibers grown onvertically aligned InAs nanowires via chemical vapour deposition” by Muhammad Arshad et al, submitted to Nanomaterial.

The authors report on the growth of gold calatalyzed InAs nanowires on InAs(111)B by CBE, followed by the growth of carbon nanofiber (CNF) by CVD on InAs substrate.

XPS is performed to understand the requirements of CNF growth on the InAs nanowire surface.

SEM images and Raman are shown to discuss the CNF growth results.

Minor : Fig1 should used (a) and (b) naming scheme, same as the rest of manuscript. Top part (without H2), contrast is overexposed, surface details are not visible.

Minor : Fig2 should label the XPS lines directly on the figure. Currently the details of the line are not even in the legend but in the main text (and thus hard to find).

=> please assign and differentiate InAs and oxide related line directly on Fig2.

Minor : Fig2 “as-grown” InAs is misleading. While we may expect C content due to TBAs and TMI during the InAs nanowire growth, oxygen-related peaks are likely to form after growth when the sample is exposed to atmosphere.

=> the authors should state clearly in their method that the InAs sample has been exposed to air (and oxidized) during the transfer from the CBE to the CVD growth chamber.

Major : the authors needs to include a schematic of their structure showing the InAs substrate, then Au+InAs nanowire, then (air) oxidation, and then branching out for substrate pre-treatment “w/o H2”, followed by another branching out for “w/o” Fe catalyst.

Currently the progression and conclusion are hard to follow. The presence of Au at the top of the InAs nanowire needs to be clearly indicated on this schematic at all steps.

Major : the author need to show the C and O XPS signal in Fig2. This is essential to understand the role of H2. Currently the discussion of the role of the pre-treatment is at best incomplete

- without H2 during pre-treatment = no Carbon & no CNF

- without H2 during growth = Carbon & CNF

- with H2 during growth = CNF

==> the fact that H2 is required for CNF growth suggests that something else than the proposed “surface cleaning” is required for CNF growth. This hypothesis is further supported by the fact that CNF growth seems to occur close to the Au-catalyst region (of InAs nanowire) and not all over the exposed InAs(111B) surface or InAs nanowire sidewalls, even when Fe is evaporated (over all surfaces).

From this perspective, H2 is blocking the CNF growth on InAs surface (not enhancing it), even in the presence of Fe.

Growth with H2+C2H2 directly at high temperature (520°C) on  "as-grown" (=oxidized) InAs nanowire directly at high temperature may bypass the pre-treatment requirements.  It is not clear whether this experiment has been performed or not.

Major : Fig3 needs to be rework to group and show the sample in a logical manner. (e)(f) being zoomed of (b) while being located after (c) and (d) is puzzling. Currently I don’t understand the logic behind its construction.

→ indicate directly on figure the important information (H2/no H2, Fe/no Fe, growth T°C, etc)

→ see also major comment relative to Table 1 & 2, as it is likely to reshape Fig3 as well.

Major : please provide zoom of CNF grown with Fe particle, as they seems to have a different morphology and length. Coordinate the presentation of these zoomed up views with the current zoom up views of Fig3 (e)(f) so that there is a clear and simple logic. Length of CNF (in addition to their presence) should be indicated in Table 1 and 2

Major : Fig4. To be compared easily by the reader, SEM images should have the same scale (top and bottom) and that scale should be clearly indicated on the figure. This is not for the reader to rescale the figure. Not to the standard of scientific publication.

Major : the meaningful results of Table 1&2 should be assembled graphically by presenting the SEM images (at same scale) of the selected samples. Currently Fig3 is an incomplete attempt at this presentation. I see several ‘coherent’ series ( C2H2 increasing Table1) and H2 variation (Table2) which are insufficiently discussed in the main text (condition being described as “optimal” without much SEM support)

Minor : Fig6 XPS need to index the lines on the figure, same as requested for Fig2

Other: given the clear affinity of CNF to the Au catalyst region of the InAs nanowire, I would have expected at least a TEM+EDX analysis of such region, possibly comparing with/without Fe. In retrospect the surface sensitive XPS feels like the wrong tool.

Fig3(c) shows strange structure, is the Au catalyst moving during the CNF growth ? Remains at the base of the CNF (i.e. growth is CNT-like) or moves at the top (VLS-like ?)

→ the main ingredient of the actual CNF growth (Au) is not really discussed and yet I feel this should have been the topic of the paper (given its title).

Conclusion : the manuscript is showing that CNF can be grown on InAs. The important conclusion (Au-related growth of CNF) is not picked up by the lengthy XPS analysis (which only shows the InAs surface…. without CNF growth), so that a large part of the discussion feels misdirected.

----

Major revisions are recommended so that the authors can better present their results into coherent experimental series.

Author Response

We thank the referee for the valuable comments, which have helped us to improve the quality of our manuscript. 

Minor : Fig1 should used (a) and (b) naming scheme, same as the rest of manuscript. Top part (without H2), contrast is overexposed, surface details are not visible.

We followed the suggestion of the referee and improved the image quality as much as possible. Unfortunately we cannot take new images because the samples got damaged accidentally.

Minor : Fig2 should label the XPS lines directly on the figure. Currently the details of the line are not even in the legend but in the main text (and thus hard to find).

=> please assign and differentiate InAs and oxide related line directly on Fig2.

We followed the suggestion of the referee - see Fig.2 of the revised manuscript.

Minor : Fig2 “as-grown” InAs is misleading. While we may expect C content due to TBAs and TMI during the InAs nanowire growth, oxygen-related peaks are likely to form after growth when the sample is exposed to atmosphere.

 => the authors should state clearly in their method that the InAs sample has been exposed to air (and oxidized) during the transfer from the CBE to the CVD growth chamber.

We followed the suggestion of the referee and replaced as-grown by  as received and labeled the components in the figure and in the caption. We also added the exposure to air (which was already mentioned in the original manuscript) to the new figure 9 which summarizes all results.

Major : the authors needs to include a schematic of their structure showing the InAs substrate, then Au+InAs nanowire, then (air) oxidation, and then branching out for substrate pre-treatment “w/o H2”, followed by another branching out for “w/o” Fe catalyst.

Currently the progression and conclusion are hard to follow. The presence of Au at the top of the InAs nanowire needs to be clearly indicated on this schematic at all steps.

We added a figure that summarizes the different growth conditions and results as requested by the referee, this is now Figure 9 in the revised manuscript. Moreover we added two sentences to clarify how we know that the nanowires are topped by Au and where we clearly state that without pretreatment there is no CVD growth (neither carbon nanoparticles nor carbon nanofibers).

Major : the author need to show the C and O XPS signal in Fig2. This is essential to understand the role of H2. Currently the discussion of the role of the pre-treatment is at best incomplete

- without H2 during pre-treatment = no Carbon & no CNF

- without H2 during growth = Carbon & CNF

- with H2 during growth = CNF

==> the fact that H2 is required for CNF growth suggests that something else than the proposed “surface cleaning” is required for CNF growth. This hypothesis is further supported by the fact that CNF growth seems to occur close to the Au-catalyst region (of InAs nanowire) and not all over the exposed InAs(111B) surface or InAs nanowire sidewalls, even when Fe is evaporated (over all surfaces).

From this perspective, H2 is blocking the CNF growth on InAs surface (not enhancing it), even in the presence of Fe. 

Growth with H2+C2H2 directly at high temperature (520°C) on  "as-grown" (=oxidized) InAs nanowire directly at high temperature may bypass the pre-treatment requirements.  It is not clear whether this experiment has been performed or not. 

We think that showing the C and O spectra does not add to the discussion - what is important, is what happens to the nanowires. As we conclude from the discussion of Figure 8 of the revised manuscript (Figure 6 of the original manuscript), the fact that CNF growth is observed only on the tips of the InAs nanowires is due to the presence of Au only there. We tried to extract information about the etching effect of hydrogen, which is responsible for the absence of carbon nanoparticles when hydrogen is present in the CVD gas mixture, but unfortunately our XPS resolution was not sufficient to deduce anything. The only thing we know is that without hydrogen pretreatment, there is no CVD growth as we already mentioned. 

Major : Fig3 needs to be rework to group and show the sample in a logical manner. (e)(f) being zoomed of (b) while being located after (c) and (d) is puzzling. Currently I don’t understand the logic behind its construction.

→ indicate directly on figure the important information (H2/no H2, Fe/no Fe, growth T°C, etc)

→ see also major comment relative to Table 1 & 2, as it is likely to reshape Fig3 as well.

We followed the suggestion of the reviewer and replaced the original Figure 3 by a new one with the details of the synthesis in each case and added a new figure (Figure 4 in the revised manuscript) to show the detail in a way that avoids confusion. 

Major : please provide zoom of CNF grown with Fe particle, as they seems to have a different morphology and length. Coordinate the presentation of these zoomed up views with the current zoom up views of Fig3 (e)(f) so that there is a clear and simple logic. Length of CNF (in addition to their presence) should be indicated in Table 1 and 2

The new figure 3 replies to these comments. One can clearly see in Fig. 3 A c that the morphology is slightly different. The same is seen in the detailed views in the new Figure 4 of the revised manuscript, as we now also indicate in the text. “which are shown in Figure 4 (a) and (b), one clearly distinguishes the nucleation of CNFs (a) and the establishment of a CNF bridge (b) between…”

Instead of the tables 1 and 2 we added Figure 5, which details the optimisation of the growth process. 

Major : Fig4. To be compared easily by the reader, SEM images should have the same scale (top and bottom) and that scale should be clearly indicated on the figure. This is not for the reader to rescale the figure. Not to the standard of scientific publication.

We modified Fig. 4 of the original manuscript (now Figure 6 in the revised manuscript) following the indications of the reviewer.

Major : the meaningful results of Table 1&2 should be assembled graphically by presenting the SEM images (at same scale) of the selected samples. Currently Fig3 is an incomplete attempt at this presentation. I see several ‘coherent’ series ( C2H2 increasing Table1) and H2 variation (Table2) which are insufficiently discussed in the main text (condition being described as “optimal” without much SEM support)

We followed the suggestion of the reviewer: instead of the tables 1 and 2 we added figure 5 which details the optimisation of the growth process.

Minor : Fig6 XPS need to index the lines on the figure, same as requested for Fig2

In order not to make the figure too crowded, we added the explanation of the color code of the components in the text and modified the figure caption to bring this to the reader’s attention.

Other: given the clear affinity of CNF to the Au catalyst region of the InAs nanowire, I would have expected at least a TEM+EDX analysis of such region, possibly comparing with/without Fe. In retrospect the surface sensitive XPS feels like the wrong tool.

Fig3(c) shows strange structure, is the Au catalyst moving during the CNF growth ? Remains at the base of the CNF (i.e. growth is CNT-like) or moves at the top (VLS-like ?)

Unfortunately we do not have TEM+EDX, however, from the new Figure 4 in the revised manuscript it is clear that the growing nanowire (a) does not show a different contrast at the free end, which would hint to Au at the tip. Therefore one can conclude that the Au is only at the top of the InAs nanowires and does not move during CNF growth.

→ the main ingredient of the actual CNF growth (Au) is not really discussed and yet I feel this should have been the topic of the paper (given its title).

The role of gold is discussed in the text from fig. 3 onwards and especially detailed in the discussion of figure 8 of the revised manuscript, where we amply consider which compounds form and how the catalytic role of Au can be understood. We respectfully disagree with the referee on the catalytic role of Au being the main topic. What is important here, is the fact that one can connect InAs nanowires with CNFs. 

Conclusion : the manuscript is showing that CNF can be grown on InAs. The important conclusion (Au-related growth of CNF) is not picked up by the lengthy XPS analysis (which only shows the InAs surface…. without CNF growth), so that a large part of the discussion feels misdirected.

The discussion of Figure 8 of the revised manuscript discusses all the details of the Au-related growth of CNF on the InAs nanowires and the new figure 9 summarizes what happens at all stages of the different protocols tested.  

Reviewer 2 Report

Comments and Suggestions for Authors

The article describes the CVD growth of carbon nanofibers on the top of InAs nanowires. This approach to formation of hybrid structures is of interest for developing energy-efficient integrated circuits. The article can be published with minor corrections. The main remark concerns to the term "nanofibers'. The tibular objects grown by the authors have diameter of several nm and their nanofiber origin was determined on the basis ID/IG ratio in Raman spectrum. I believe that it is not sufficient for discrimination of nanofibers and nanotubes and probably the authors synthesize highly defective carbon nanotubes. Anyway some consideration of this issue is necessary. Besides of that the structural fetures of the objects synthesized (diameter, length and spread of those)  should be determined more exactly.

Useless hyphens like in “de- composition”  (p. 10) and others should be avoided.

Comments on the Quality of English Language

The English Language is noy perfect and shold be corrected

Author Response

The article describes the CVD growth of carbon nanofibers on the top of InAs nanowires. This approach to formation of hybrid structures is of interest for developing energy-efficient integrated circuits. The article can be published with minor corrections. The main remark concerns to the term "nanofibers'. The tibular objects grown by the authors have diameter of several nm and their nanofiber origin was determined on the basis ID/IG ratio in Raman spectrum. I believe that it is not sufficient for discrimination of nanofibers and nanotubes and probably the authors synthesize highly defective carbon nanotubes. Anyway some consideration of this issue is necessary. Besides of that the structural fetures of the objects synthesized (diameter, length and spread of those)  should be determined more exactly.

As we explain in the discussion of the Raman data, if single-walled CNTs were formed, one would expect to observe the radial breathing mode (RBM) signature, which is not seen in the spectrum. The G peak at 1592 cm-1 is characteristic of sp2 carbon, and the strong D peak at 1368 cm-1 points to defects and disordered graphitic material. The high ID/IG ratio of 0.75 corresponds to a low degree of order, typical for CNFs. As we show in the SEM images, the length and the diameter of the CNFs varies: without hydrogen in the CVD gas mixture the CNFs are very short, with the iron catalyst present the CNFs are significantly longer and thinner than those obtained when growing without Fe. We believe that the new scheme in Figure 9 of the revised manuscript, which summarises the results obtained in the various growth conditions, makes this more evident to the reader.

Useless hyphens like in “de- composition”  (p. 10) and others should be avoided.

We eliminated this and other typos in the revised version of the manuscript.

Reviewer 3 Report

Comments and Suggestions for Authors

1.       The adoption of C2H2 is to lower the growth temperature. Why not use a solid precursor (e. g. camphor, Materials Letters 218 (2018) 90–94) since the growth temperature can be further lowered.

2.       Nowadays, 3D-graphene can be deposited on most substrates. The unique interconnecting morphology of 3D-graphene leads to a high light-harvesting ability without sacrificing its electrical property (IEEE ELECTRON DEVICE LETTERS, VOL. 43, NO. 11, 2022). So compared with 3D-graphene, what are the advantages and disadvantages of carbon nanofibers?

Author Response

Replies to Reviewer 3

  1. The adoption of C2H2 is to lower the growth temperature. Why not use a solid precursor (e. g. camphor, Materials Letters 218 (2018) 90–94) since the growth temperature can be further lowered. 

Using a solid precursor is beyond the scope of this paper where we explored CVD growth with acetylene with and without hydrogen. As evident from Figure 5(b) of the revised manuscript, 525 oC is the optimal growth temperature when working with the optimized gas fluxes: C2H2 3.6 sccm and H2 2.0 sccm, both with and without the iron catalyst.

  1.  Nowadays, 3D-graphene can be deposited on most substrates. The unique interconnecting morphology of 3D-graphene leads to a high light-harvesting ability without sacrificing its electrical property (IEEE ELECTRON DEVICE LETTERS, VOL. 43, NO. 11, 2022). So compared with 3D-graphene, what are the advantages and disadvantages of carbon nanofibers?

InAs nanowires cannot be grown on a graphene surface, and we do not understand how two InAs nanowires could be connected with a graphene nanoribbon on the InAs(111)B substrate (we could not find any paper demonstrating that graphene can be grown on this surface), so we do not see how the alternative suggested by the reviewer could be realised.

Reviewer 4 Report

Comments and Suggestions for Authors

In their paper "Growth and characterization of carbon nanofibers ...", Muhammad Arshad and coworakers present the preparation and characterization of carbon nanofibers, pependicular to vertically aligned InAs nanowires. Mainly through Raman scattering, the authors identify the chemical nature of the individual components. The methodology of preparation  of the model system seems scientifically sound. Fabrication of such connecting nanofibers by CVD is highly innovative and definitely of interest for the readers of "nanomaterials".

However, some points need to be corrected before the paper is ready for publication.

1. page 5, last paragraph: The authors should quantify the growth of connecting nanofibers. Does this happen only once or twice, or is it more frequent?

2. page 6, last paragraph: It should be "Table 2", not "Ta- ble 2".

3. Tables 1 and 2: The authors should consider moving these tables to the supporting information. They should also provide an estimate of the error for the last column, instead of using "to".

4. Figure 4: What are the ratios of CNF/InAs NW for the two samples?

5. Figure 6: Please recheck this Figure. The order does not match the sequence described in the caption or in the text.

6. page 10, last sentence: "Our results demonstrate that controlled interconnections between adjacent InAs NWs with carbon fibers can be obtained via CVD." The authors should reformulate this sentence. It appears that the growth of CNFs perpendicular to the InAS NWs is controlled, while the interconnections are random.

Author Response

1. page 5, last paragraph: The authors should quantify the growth of connecting nanofibers. Does this happen only once or twice, or is it more frequent?

The SEM images of Figure 6 of the revised manuscript clearly show that when no Fe is present and only 15% of the InAs nanowires is decorated by CNFs, connection of two nanowires is a rare event while when Fe is present and CNFs have grown on about half of the InAs nanowires, the connection happens much more frequently.    

2. page 6, last paragraph: It should be "Table 2", not "Ta- ble 2".

We eliminated similar typos in the revised version of the manuscript.Since we replaced the tables by Figure 5, "Table 2" no longer appears in the revised manuscript.

3. Tables 1 and 2: The authors should consider moving these tables to the supporting information. They should also provide an estimate of the error for the last column, instead of using "to".

We replaced the tables by Figure 5 following the suggestion of another reviewer.

4. Figure 4: What are the ratios of CNF/InAs NW for the two samples?

This figure has become Figure 6 in the revised manuscript. We now specify in the figure caption that Figure 6(a) is the sample already shown in Figure 3 (c) but on a larger scale and Figure 6(b) is the sample already shown in Figure 3 (b) but on a larger scale, so that it is evident to the reader that with  Fe 50% of the InAs nanowires show CNF growth and without Fe only 15% of the InAs nanowires show CNF growth. We repeat this information also in Figure 9, which summarises the results obtained in all different growth conditions.

5. Figure 6: Please recheck this Figure. The order does not match the sequence described in the caption or in the text.

We thank the reviewer for alerting us to this mistake, which we corrected in the revised manuscript.

6. page 10, last sentence: "Our results demonstrate that controlled interconnections between adjacent InAs NWs with carbon fibers can be obtained via CVD." The authors should reformulate this sentence. It appears that the growth of CNFs perpendicular to the InAS NWs is controlled, while the interconnections are random.

We reformulated the sentence as suggested by the reviewer: "Our results demonstrate that interconnections between adjacent InAs NWs with carbon fibers can be obtained via CVD, however to which of the neighboring NWs a connection is made is random."

Reviewer 5 Report

Comments and Suggestions for Authors

This  work presents a method for the controlled growth of carbon nanofibers (CNFs) on vertically aligned indium arsenide (InAs) nanowires. The results obtained demonstrate a compelling example of controlled intercon-nections between adjacent InAs nanowires and carbon fibers. Some issues should be addressed before publication:

(1) Page 7, Table 1 and 2, the Table questions are suggested to be placed before the table. Otherwise, the Tables are suggested to be shown in three-wire meter form;

(2)Page 8, "Figure 5 displays the Raman spectra acquired on the InAs substrate covered with the as-grown InAs NWs, after annealing at 430 °C, and after CVD growth as described in......", the first letter needs a space;

(3) Page 9, "Additionally, the formation of iron compounds with indium and/or arsenide is unlikely at our growth temperatures", the aurhors are suggested explain it more detaily;

(4) Page 10, "Neverthelss, it is worth noting that the AuIn2 alloy in form of nanoparticles has prevously been found to catalyse the growth of InAs na-notrees even at very low temperature [15] and it may be responsible for the CNF synthesis in our case. The nanometric size of the Au-In wires in the samples studied here can further enhance their reactivity, as already observed in several nanostructured materials. [37, 38,39,40]", the authors are suggested to provide more evidence.

Overall, the paper is well organised and can be accepted after minor reversion.

Author Response

(1) Page 7, Table 1 and 2, the Table questions are suggested to be placed before the table. Otherwise, the Tables are suggested to be shown in three-wire meter form;

Prompted by another reviewer, we replaced the tables with Figure 5 in the revised manuscript so this comment is no longer relevant. 

(2)Page 8, "Figure 5 displays the Raman spectra acquired on the InAs substrate covered with the as-grown InAs NWs, after annealing at 430 °C, and after CVD growth as described in......", the first letter needs a space;

We added the tab at the beginning of this paragraph.

(3) Page 9, "Additionally, the formation of iron compounds with indium and/or arsenide is unlikely at our growth temperatures", the aurhors are suggested explain it more detaily;

In the original manuscript we cited two references, 30 and 31, referring to the phase diagrams of FeSe and CoIn, two alloys that are similar to the compounds discussed here. These references show that no alloys are formed at the temperatures we use in CVD growth. From this we conclude that  " ...the formation of iron compounds with indium and/or arsenide is unlikely at our growth temperatures".

(4) Page 10, "Neverthelss, it is worth noting that the AuIn2 alloy in form of nanoparticles has prevously been found to catalyse the growth of InAs na-notrees even at very low temperature [15] and it may be responsible for the CNF synthesis in our case. The nanometric size of the Au-In wires in the samples studied here can further enhance their reactivity, as already observed in several nanostructured materials. [37, 38,39,40]", the authors are suggested to provide more evidence.

The two sentences which precede those cited by the reviewer read "On the other hand, Indium has a high solubility in Au, and readily forms Au-In alloys. [36] Unfortunately, due to the low Au concentration in our sample, we were not able to distinguish the formation of an In-Au alloy, and the In3d spectra acquired before and after the CVD process (Figure 8 (c)) do not show any significant differences." These sentences explain why we cannot say more about an In-Au alloy and can only speculate that such a compound may be responsible for the CNF synthesis in our case.

Round 2

Reviewer 3 Report

Comments and Suggestions for Authors

I have not found any response to my questions in the revised version. I suggest at least in the introduction portion, the comments (and necessary references) on the following questions should be supplemented.

1. The adoption of C2H2 is to lower the growth temperature. Why not use a solid precursor (e. g. camphor, Materials Letters 218 (2018) 90–94) since the growth temperature can be further lowered.

2. Nowadays, 3D-graphene can be deposited on most substrates. The unique interconnecting morphology of 3D-graphene leads to a high light-harvesting ability without sacrificing its electrical property (IEEE ELECTRON DEVICE LETTERS, VOL. 43, NO. 11, 2022). So compared with 3D-graphene, what are the advantages and disadvantages of carbon nanofibers?

Author Response

We had replied to both comments of the referee but since our response does not seem to have satisfied the reviewer, we elaborate more on our response: Using a solid precursor is beyond the scope of this paper where we explored CVD growth with acetylene. As evident from Figure 5 of the revised manuscript, 525 oC is the optimal growth temperature when working with the optimized gas fluxes: C2H2 3.6 sccm and H2 2.0 sccm and employ the iron catalyst. A gaseous precursor is preferable to a solid one in view of a possible future development towards up-scaling since all industrial production uses gaseous precursors whenever possible so because exchange of precursor reservoirs is easier (exchange of gas bottle in place of exchange of crucible with solid precursor). Moreover, also the argument of growth at lower temperature is not justified since the reference given reports on growth of graphene on Ni foam at 550 oC while we grow nanofibers to connect InAs nanowires at 525 oC. The materials are different and our growth temperature is lower. We therefore respectfully disagree with the suggestion to include a sentence about solid precursors in the introduction.

2. Graphene is not relevant for the scope of our study because although InAs nanowires can be grown on graphene (Kang, J.-H., Ronen, Y., Cohen, Y., Convertino, D., Rossi, A., Coletti, C., Heun, S., Sorba, L., Kacman, P., Shtrikman, H.; Semiconductor Science and Technology (31); 115005 (2016) 10.1088/0268-1242/31/11/115005), this does not lead to connection of single couples of nanowires as we attempted in this work. Moreover, we do not see how two nanowires could be connected with a graphene nanoribbon on the InAs(111)B substrate (we could not find any paper demonstrating that graphene can be grown on this surface), so we do not see how the alternative suggested by the reviewer could be realized. This is why we do not refer to graphene growth in our introduction.